# Image Findings as Predictors of Fall Risk in Patients with Cerebrovascular Disease

**DOI:** 10.3390/brainsci13121690

**Published:** 2023-12-07

**Authors:** Tatsuya Tomita, Hisanori Yuminaga, Hideki Takashima, Takashi Masuda, Tomoo Mano

**Affiliations:** 1Department of Rehabilitation, Nara Prefectural General Medical Center, Nara 630-8054, Japan; graduation0615@gmail.com (T.T.); takashimapt@gmail.com (H.T.); amenouo@hotmail.com (T.M.); 2Department of Physical Therapy, Kansai Vocational College of Medicine, Osaka 558-0011, Japan; yuminaga@kansai.ac.jp

**Keywords:** cerebrovascular disorders, fall risk, lateral ventricular enlargement, periventricular lucency, thalamus, interior capsule

## Abstract

This study examined computed tomography findings in patients with cerebrovascular disease and determined predictors for falls. Images of the head were divided into 13 regions, and the relationships between computed tomography findings and the presence or absence of falls were investigated. A total of 138 patients with cerebrovascular disease (66% men, aged 73.8 ± 9.6 years) were included. A comparison between the fall and non-fall groups revealed a significant difference in the total functional independence measure scores and imaging findings at admission. Logistic regression analysis showed that the thalamus (*p* < 0.001), periventricular lucency (*p* < 0.001), lateral hemisphere room enlargement (*p* < 0.05), and age (*p* < 0.05) were related to the presence or absence of falls. For the 42 patients with cerebral hemorrhage, the thalamus (*p* < 0.01), periventricular lucency (*p* < 0.05), lateral ventricle vicinity (*p* < 0.05), and posterior limb of the internal capsule (*p* < 0.05) were extracted as factors related to the presence or absence of falls. For the 96 patients with cerebral infarction, the thalamus (*p* < 0.001), periventricular lucency (*p* < 0.01), and anterior limb of the internal capsule (*p* < 0.05) were extracted as factors related to the presence or absence of falls. This study found a relationship between the thalamus, lateral ventricle enlargement, periventricular lucency, and falls. Fall prognosis can potentially be predicted from computed tomography findings at admission.

## 1. Introduction

Falls have a major impact on the average lifespan of older people. Older adults are reported to experience falls at least once per year [1,2]. Many hospitals have implemented measures to prevent falls among hospitalized patients [3]. Multiple definitions of falls exist in the literature [4]. Major risk factors for falls, which can occur at home and during hospitalization, include cognitive dysfunction, motor impairment, and loss of balance function [5,6].

Many patients are hospitalized because of cerebrovascular disorders, which refers to disruption in the cerebral blood vessels, resulting in neurological abnormalities such as impaired consciousness and hemiplegia. Cerebrovascular disorders cause motor dysfunction and postural adjustment disorders, increasing the risk of falls. Falls in patients with cerebrovascular disease not only cause bleeding and fractures but also affect rehabilitation and quality of life. Therefore, predicting and preventing falls is important to prevent injuries and improve quality of life. Previous studies report an association between basal ganglia injuries and falls [7] and show that brain lesions affect gait. However, specific brain lesions affecting gait recovery have not yet been identified [8]. In previous research, brain imaging findings have been shown to predict physical function; however, no association between these findings and falls has been reported.

Various tools are used to assess patient falls [3]. Most assessments of fall risks are based on physical and cognitive function, medication, and other factors. Additionally, these tools often focus on assessing post-fall evaluations rather than fall prediction. Many studies use the Time Up Go test (TUG) to assess fall risk [9,10,11]. However, because TUG is strongly influenced by motor function, an evaluation combined with cognitive function, such as dual-task testing, is necessary [12]. Additionally, walking ability is impaired in 80% of patients after stroke [13]. TUG, which assesses fall risk by walking, may not be possible for stroke patients with decreased walking ability. Therefore, predicting falls based on factors other than motor function is warranted. Although previous studies have mostly evaluated motor function, we believe that factors other than motor function may contribute to fall risk [7,8]. Brain imaging findings may be related to walking ability and physical function. Some brain imaging findings are associated with reduced walking ability [13]. Such findings can also be caused by falls. However, little is still known about using computed tomography (CT) for fall prediction. The purpose of this research is to identify a method that can be used to easily predict falls. Falls worsen the quality of life of older people and increase the risk of becoming bedridden [14]. We hypothesized that the cause of falls can be predicted from brain imaging findings. This study examined CT findings in patients with cerebrovascular disease and determined factors as predictors for falls.

## 2. Materials and Methods

### 2.1. Participants and Methods

We enrolled patients with cerebrovascular disease who were admitted to Yoshieikai Hospital between January 2016 and October 2019 and met the inclusion criteria. The major inclusion criteria were as follows: (1) 20–90 years old and (2) the patients provided informed consent. The major exclusion criteria were as follows: (1) patients without brain imaging findings, (2) patients with impaired consciousness, and (3) patients who were judged by the researchers as unsuitable for participation in this study. Written informed consent was obtained from all patients. All study procedures were performed in accordance with the ethical standards of the Institutional Research Committee and adhered to the principles of the Declaration of Helsinki and Ethical Guidelines for Medical and Health Research Involving Human Subjects in Japan. All data were collected from medical records. In this study, we defined fall using Gibson’s definition [4]. Gibson’s definition is as follows: “[a fall] is the inadvertent fall of a person onto the same or lower plane, not due to external force caused by another person, loss of consciousness, sudden onset of paralysis, epileptic seizure, etc”. Fall history included the number of falls during hospitalization.

### 2.2. Clinical Assessments

#### 2.2.1. CT

The analysis was based on a previous study [15] in which the lesion in the head CT was divided into three sections: above the fourth ventricle level, at the third ventricle level, and at the midventricular level. The anatomical regions were divided into the basal forebrain and midbrain, thalamus, putamen, anterior limb of the internal capsule, posterior limb of the internal capsule, anterior parietal lobe, middle parietal lobes, posterior parietal lobe adjacent to the lateral ventricle, and distant from the lateral ventricle. In addition, the presence or absence of lateral ventricular enlargement and periventricular lucency (PVL) was confirmed, and 13 brain sites, including the regions mentioned above, were examined. PVL was visually confirmed, and lateral ventricular enlargement was confirmed using the Evans index as a lesion size ≥0.3. Regions with lesions were assigned 1 point, and those without lesions were assigned 0 points (Figure 1). We defined the presence or absence of falls based on Gibson’s definition (described above) [4].

#### 2.2.2. Functional Independence Measure (FIM)

FIM measurements were performed by evaluating and measuring subjects on a scale of 0–7. FIM can be evaluated from independence to total assistance.

#### 2.2.3. Mini-Mental State Examination (MMSE)

MMSE diagnoses cognitive functions, such as time and place orientation, immediate recall and delayed reproduction, and numeracy, language, and graphic ability. The MMSE contains 11 items, and the perfect score is 30 points. A higher score indicates higher cognitive function. An MMSE score lower than 23 points indicates dementia.

#### 2.2.4. Functional Ambulation Categories (FAC)

FAC uses a walking path or staircase of about 15 m.

It measured on a scale of 0–6, from inability to walk to independence. It does not matter whether or not prosthetics are used.

### 2.3. Statistical Analyses

Participants were divided into fall and non-fall groups based on age, FIM, and MMSE score at admission. FAC and duration from onset to CT were compared using a two-group unpaired *t*-test. We performed an unadjusted model and a separate model adjusted for age in the regression analyses. The objective variable was the number of falls, and the explanatory variables were the findings from the 13 regions identified in the images. Furthermore, the patients were classified into cerebral infarction and hemorrhage groups based on the CT findings.

Considering the possibility of multicollinearity, we computed the variance inflation factors (VIF) between variables. All significance levels were set at *p* < 0.05. Statistical analyses were performed using the statistical software R ver.2.71 (R Foundation, Vienna, Austria).

## 3. Results

### 3.1. Patient Characteristics

A total of 138 patients with cerebrovascular disease (91 men, 47 women, aged 73.8 ± 9.6 years; 42 with cerebral hemorrhage, 96 with cerebral infarction) were included. They were divided into the fall (n = 43) and non-fall (n = 95) groups.

No significant differences in age, MMSE, and FAC at admission between the fall and non-fall groups were observed, but significant differences existed in the total FIM score and total CT findings at admission (Table 1).

### 3.2. Logistic Regression Analysis

Logistic regression analysis was performed on 138 patients. We performed an unadjusted model (Table 2) and a separate model adjusted for age in the regression analyses (Table 3). The thalamus (odds ratio [OR]: 15.6, 95% confidence interval [CI]: 5.78–42.0, *p* = 0.00000005), PVL (OR: 6.45, 95% CI: 2.25–18.5, *p* = 0.0005), lateral ventricular enlargement (OR: 2.31, 95% CI: 1.09–4.9, *p* = 0.029), and age (OR: 0.94, 95% CI: 0.89–0.99, *p* = 0.03) were extracted as factors related to the presence or absence of falls. We computed VIF to confirm the independence of the explanatory variables and found that each variable had a VIF of <10; thus, no multicollinearity was determined (Table 3).

### 3.3. Logistic Regression Analysis Classified into Cerebral Hemorrhage and Cerebral Infarction

We analyzed the factors contributing to falls in patients with cerebral hemorrhage and cerebral infarction separately. In the cases with cerebral hemorrhage, the thalamus (OR: 241, 95% CI: 7.0–8280, *p* = 0.002), PVL (OR: 30.9, 95% CI: 1.57–607, *p* = 0.02), the region adjacent to the lateral ventricle (OR: 6.38, 95% CI: 1.32–30.8, *p* = 0.02), and the posterior limb of the internal capsule (OR: 9.79, 95% CI: 1.65–58.1, *p* = 0.012) were factors related to the presence or absence of falls (Table 4).

In the cases of cerebral infarction, the thalamus (OR: 10.1, 95% CI: 2.91–34.8, *p* = 0.0002), PVL (OR: 6.13, 95% CI: 1.66–22.6, *p* = 0.006), and the anterior limb of the internal capsule (OR: 4.43, 95% CI: 1.11–17.8, *p* = 0.03) were extracted as factors related to the presence or absence of falls (Table 5). When VIF was evaluated to confirm the independence of explanatory variables for cerebral hemorrhage and cerebral infarction cases, it was shown that the VIF of each variable was ≤10.

## 4. Discussion

This study suggested that fall risk can be easily assessed using CT images without requiring patients to walk. We believe that if we can understand the risk of falling before a physical examination, we may be able to better prevent falls. The thalamus, PVL, and lateral ventricle enlargement were extracted as factors related to the presence or absence of falls. Patients with lesions in these areas are at risk of falling; therefore, brain imaging findings may improve fall risk assessment systems.

Damage to the radial crown and posterior limb of the internal capsule can affect motor recovery and functional outcome. Damage to the putamen and globus pallidus is associated with poor recovery in patients with stroke [16]. Furthermore, a strong correlation exists between activities of daily living independence and balance function in patients with cerebrovascular disease [17,18,19]; therefore, balance function is an important predictor of falls. This suggests a relationship between disorders affecting body parts that cause a decline in motor and balance functions. Falls occur when a mismatch between physical function and environmental information is detected and where an association with a decline in attentional function is evident. Thus, examining the mechanism of falls from various aspects is necessary.

Here, the thalamus, PVL, and lateral ventricle enlargement were identified as factors related to the presence or absence of falls. The thalamus, PVL, and forelimb of the internal capsule were identified as factors associated with the presence or absence of falls in cerebral infarction. In cerebral hemorrhage, the thalamus, PVL, the region adjacent to the lateral ventricle, and the posterior limb of the internal capsule were identified as factors related to the presence or absence of falls.

The thalamus controls important functions associated with sensory information processing and motor regulation and is considered the highest center in the brain, leading to decreased arousal. The ascending reticular activating system is involved in the thalamus and a network connecting the brainstem reticular formation, nonspecialized thalamic nuclei, and parts of the cerebrum. The system is considered to play a role in regulating consciousness [20,21]. PVL is identified as a faint low density around the ventricles; it refers to changes due to ischemia in the white matter and can be evident even in healthy older people. In CT, it is written as PVL [22]. The area where PVL is likely to occur corresponds to the deep borders of the anterior, middle, and posterior cerebral arteries. Therefore, ischemia is also likely to occur. The anterior choroid plexus arteries, which are distributed around the lateral ventricles, perfuse various important neural tissues. The choroid plexus arteries supply the cerebral cortex, although most of the supply goes to the posterior cerebral arteries. Ischemia of the anterior choroid plexus artery impairs the supply of nutrients to the deep white matter of the cerebral cortex, resulting in decreased cerebral activity. Consequently, PVL is believed to be observed as ischemic changes in the deep white matter around the lateral ventricle. This ischemic change may indirectly affect the thalamus, which controls the posterior cerebral artery territory. Furthermore, PVL is extremely important as an aggravating factor for cognitive decline [23]. Ischemic changes in the brain impair the body’s information processing systems necessary to selectively gather information from the outside world and choose appropriate actions, increasing the risk of falls.

The enlargement of the lateral ventricles is associated with atrophic changes in the white matter [24]; the greater these changes are around the anterior horn of the lateral ventricles, the worse the upright postural control is. In this study, the sway of the center of gravity was not investigated, and any effects of the sway of the center of gravity are unclear. Considering that PVL was observed due to ischemic changes around the white matter, atrophic changes around the white matter might have occurred. Therefore, the enlargement of the lateral ventricles may be involved in falls.

Regarding the cases with cerebral hemorrhage, in addition to the thalamus and PVL, the region adjacent to the lateral ventricle and the posterior limb of the internal capsule were extracted as risk factors. These structures are thought to participate in the corticospinal tract, which affects the motor function of the lower limbs. Nerve fibers running from the cortex to the posterior limb of the internal capsule also descend near the lateral ventricle, and these lesions affect lower limb motor function. This anatomical consideration has an effect on falls. In the cases with cerebral infarction, in addition to the thalamus and PVL, the anterior limb of the internal capsule was extracted as a risk factor. Damage to the anterior limb of the internal capsule may cause higher brain dysfunction, leading to decreased decision-making capability and risk of impulsive behavior. Furthermore, the reticular activation system is involved in arousal, and we believe that decreased arousal and higher brain dysfunction affect falls. A relationship between white matter lesions and falls has been previously reported [25]. As reported in previous studies, physical function is important for preventing falls in older adults [26]. Although evaluations for fall risk in patients with cerebral vascular disease, such as TUG, Berg Balance Scale score (BBS), Stroke Assessment of Fall Risk (SAFR), and Morse Fall Scale are used in many hospitals, these evaluations do not predict falls on their own [27,28,29]. Based on these results, we believe it might be meaningful to add the evaluation of brain imaging findings to the physical function evaluation of the lower extremities when evaluating falls. The majority of the brain that was found to be associated with falls in this study is the part that affects alertness and cognitive function, and we believe that alertness and cognitive function influenced falls.

A major limitation of this study was the small sample size of 138 patients. Therefore, this study did not control for possible “covariates”. Additionally, this study did not present the main clinical characteristics of the participants (i.e., comorbidities and number of medications consumed daily). The people included in this study were known to have cerebrovascular disease, were definitely taking medications, and had some comorbidities. We think we should revisit this as we increase the number of cases of cerebrovascular disease [30]. Furthermore, it was necessary to evaluate the relationship between physical functions such as motor function, sensory function, cognitive function, and balance function. Since we also extracted areas where nerve fibers involved in motor function are present, such as the posterior limb of the internal capsule and the area adjacent to the lateral ventricle, it is possible that purely motor function is affected. We believe that physical findings remain important, as not all clinical professionals have the financial means to apply imaging tests. Hospitals and facilities are taking several steps to prevent falls, such as bed rails or restraints. We believe that by adding predictions using CT images to multi-faceted verification, we can further improve the accuracy of fall risk estimation.

## 5. Conclusions

The image-based prediction of falls will facilitate risk management and effective interventions during rehabilitation. The effect may become more accurate by combining it with physical functions.

## Figures and Tables

**Figure 1 brainsci-13-01690-f001:**
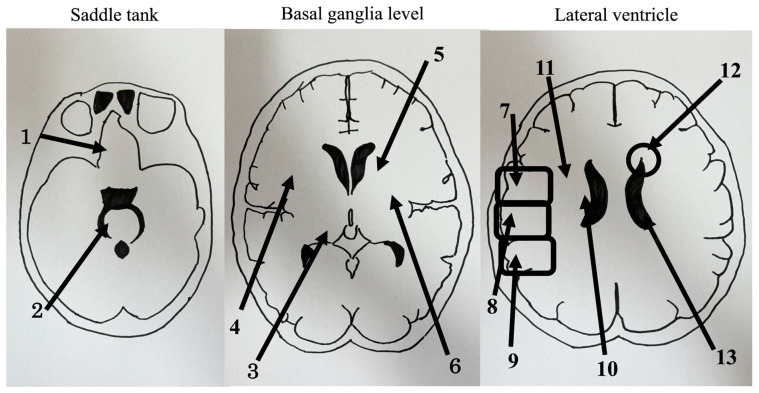
Computed tomography findings are divided into (**1**) the basal forebrain, (**2**) midbrain, (**3**) thalamus, (**4**) putamen, (**5**) anterior limb of the internal capsule, (**6**) posterior limb of the internal capsule, (**7**) anterior parietal lobe, (**8**) middle parietal lobe, (**9**) posterior parietal lobe, (**10**) adjacent to the lateral ventricle, (**11**) distant from the lateral ventricle, (**12**) periventricular lucency, (**13**) lateral ventricle.

**Table 1 brainsci-13-01690-t001:** Patient background information.

	Fall Group (n = 43)	Non-Fall Group (n = 95)	*p* Value
Age	73.6 ± 10.0	73.9 ± 9.55	0.84
	52–90 (range)	47–88 (range)	
Duration from onset to CT	36.1 ± 22.9	38.2 ± 24.4	0.63
	0–108 (range)	0–110 (range)	
MMSE	14.5 ± 9.1	14.6 ± 11.2	0.92
	0–28 (range)	0–30 (range)	
FIM	34.9 ± 12.9	43.4 ± 24.4	0.008 **
	21–119 (range)	18–123 (range)	
Total number of CT findings	2.67 ± 1.42	1.89 ± 1.47	0.004 **
	0–4 (range)	0–6 (range)	
FAC	2.0 ± 1.66	2.27 ± 2.0	0.40
	1–5 (range)	0–5 (range)	

CT: computed tomography, MMSE: Mini-Mental State Examination, FIM: functional independence measure, FAC: functional ambulation categories. ** *p* < 0.01.

**Table 2 brainsci-13-01690-t002:** Logistic regression analysis in unadjusted model.

	OR	95% CI	*p* Value	VIF
PVL	6.3	2.16–18.3	0.0007 ***	1.04
Thalamus	16.2	5.43–47.8	0.00000005 ***	1.04
Divided into the basal forebrain	0.000001	0–0	0.993	1
Lateral ventricle enlargement	1.21	0.44–3.27	0.704	1
Distant from the lateral ventricle	1.96	0.44–8.6	0.37	1.01
Adjacent to the lateral ventricle	0.61	0.21–1.71	0.3	1.01
Posterior parietal lobe	1.12	0.19–6.82	0.336	1.06
Anterior parietal lobe	1.1	0.299–6.59	0.9	1.24
Middle parietal lobe	2.79	0.53–14.7	0.22	1.28
Posterior limb of the internal capsule	2.37	0.79–7.08	0.121	1.07
Anterior limb of the internal capsule	2.04	0.52–7.97	0.3	1.03
putamen	0.41	0.14–1.22	0.11	1.04

CI: confidence interval, OR: odds ratio, PVL: periventricular lucency, VIF: variance inflation factor. *** *p* < 0.001.

**Table 3 brainsci-13-01690-t003:** Logistic regression analysis in age-adjusted model.

	OR	95% CI	*p* Value	VIF
Age	0.945	0.89–0.99	0.03 *	1.26
PVL	6.45	2.25–18.5	0.0005 ***	1.2
Thalamus	15.6	5.78–42.0	0.00000005 ***	1.09
Divided into the basal forebrain	0.000006	0–0	0.993	1
Lateral ventricle enlargement	2.31	1.09–4.9	0.029 *	1.04
Distant from the lateral ventricle	0.782	0.255–2.4	0.668	1.01
Adjacent to the lateral ventricle	1.29	0.612–2.71	0.506	1.01
Posterior parietal lobe	1.88	0.519–6.82	0.336	1.06
Anterior parietal lobe	1.1	0.299–4.06	0.884	1.24
Middle parietal lobe	1.07	0.36–3.15	0.909	1.29
Posterior limb of the internal capsule	1.76	0.798–3.9	0.161	1.08
Anterior limb of the internal capsule	1.3	0.476–3.56	0.607	1.03
Putamen	0.596	0.276–1.29	0.188	1.06

CI: confidence interval, OR: odds ratio, PVL: periventricular lucency, VIF: variance inflation factor. * *p* < 0.05; *** *p* < 0.001.

**Table 4 brainsci-13-01690-t004:** Cerebral hemorrhage logistic regression analysis.

	OR	95% CI	*p* Value	VIF
Age	0.812	0.67–0.97	0.02 *	1.24
PVL	30.9	1.57–607	0.02 *	1.42
Thalamus	241	7–8280	0.002 **	1.97
Divided into the basal forebrain				
Lateral ventricle enlargement	3.24	0.68–15.3	0.138	1.2
Distant from the lateral ventricle	0.15	0.009–2.44	0.183	1.09
Adjacent to the lateral ventricle	6.38	1.32–30.8	0.02 *	1.2
Posterior parietal lobe	1.71	0.185–15.9	0.635	1.03
Anterior parietal lobe	6.25	0.389–100	0.196	1.13
Middle parietal lobe	0.158	0.0122–2.04	0.158	1.15
Posterior limb of the internal capsule	9.79	1.65–58.1	0.012 *	1.63
Anterior limb of the internal capsule	3.6	0.19–68.5	0.394	1.04
Putamen	0.281	0.0496–1.59	0.152	1.6

CI: confidence interval, OR: odds ratio, PVL: periventricular lucency, VIF: variance inflation factor. * *p* < 0.05; ** *p* < 0.01.

**Table 5 brainsci-13-01690-t005:** Cerebral infarction logistic regression analysis.

	OR	95% CI	*p* Value	VIF
Age	0.98	0.919–1.06	0.74	1.15
PVL	6.13	1.66–22.6	0.006 **	1.03
Thalamus	10.1	2.91–34.8	0.0002 ***	1.14
Divided into the basal forebrain	0.000005	0–0	0.99	1
Lateral ventricle enlargement	1.47	0.552–3.91	0.44	1.03
Distant from the lateral ventricle	1.27	0.348–4.63	0.71	1.01
Adjacent to the lateral ventricle	0.72	0.262–1.99	0.53	1.04
Posterior parietal lobe	2.66	0.451–15.7	0.28	1.18
Anterior parietal lobe	0.21	0.025–1.73	0.14	1.54
Middle parietal lobe	3.49	0.881–13.8	0.07	1.4
Posterior limb of the internal capsule	0.36	0.098–1.37	0.13	1.26
Anterior limb of the internal capsule	4.43	1.11–17.8	0.03 *	1.28
Putamen	0.32	0.105–1.03	0.056	1.17

CI: confidence interval, OR: odds ratio, PVL: periventricular lucency, VIF: variance inflation factor. * *p* < 0.05; ** *p* < 0.01; *** *p* < 0.001.

## Data Availability

The datasets generated in this study are available upon reasonable request from the corresponding author.

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
