# Peer review of "Image Findings as Predictors of Fall Risk in Patients with Cerebrovascular Disease"

_brainsci, 2023, doi:10.3390/brainsci13121690_

Round 1
Reviewer 1 Report
Comments and Suggestions for Authors
Dear all, here are suggestions to broaden the understanding of your manuscript:
Abstract
I consider that the text met the criteria of an abstract.
Introduction
1. I suggest that the authors introduce and present what "cerebrovascular disease" is. This is essential so that readers who are not familiar with this disease can better understand the content of your study. Furthermore, the proportion of information between falls and cerebrovascular disease is unequal;
2. The authors stated that "Many studies have reported using the Time Up Go test (TUG) to assess fall risk [9]". However, only one study was presented. The literature is vast with examples, please expand the discussion;
3. After this statement "Additionally, patients with cerebrovascular disease may not be able to undergo tests such as the TUG owing to hemiplegia or muscle weakness" I suggest complementing with examples of studies that addressed the case, and can support this statement;
4. It is still necessary to present a strong "justification" for carrying out the study. For example: "To date, there is a gap in studies on ....", or "Author XXX, investigated XXX, Author XXX, evaluated XXX, however, little is still known about XXXX;
5. I suggest working with a single objective. So far there are two versions: End of the introduction: "This study used brain computed tomography (CT) findings at admission to examine factors associated with falls". And abstract: "This study examined computed tomography findings in patients with cerebrovascular disease and determined factors as predictors for falls".
Method
1. I suggest a better organization of this section, creating subsections:
a) Presentation of participants;
b) Inclusion and exclusion criteria: detail how the history of falls was collected;
c) Methods used to collect data, specifying the instruments (devices/machines);
c1) Authors must mention the use of the Mini-Mental State Examination (MMSE) instrument in this subsection;
d) Statistical procedures
* I consider a weakness of the present study that the analyzes were not conducted controlling possible "covariates". It is known that the individuals included have illnesses and certainly consumed medications, as well as having some comorbidities. Therefore, it would be interesting if the regression analyzes presented an unadjusted OR model and another OR adjusted for confounding factors.
Discussion
1. I consider that the authors showed ownership in this section;
2. On the other hand, the issue of limitations requires a "strong formulation on the "Covariates" point, as well as a failure to present the main clinical characteristics of the participants (i.e., comorbidities, number of medications consumed daily);
3. I suggest, presenting strengths of the study;
4. I suggest presenting suggestions for future investigations: directions based on the results found.
Conclusion
1. I consider it clear and consistent.
2. point for reflection: The authors stated the following "Image
based prediction of falls will facilitate risk management and effective interventions during rehabilitation". However, it is known that not all clinical professionals have the financial means to apply imaging tests to detect risk of falls in their patients. Therefore, I suggest that this point be presented in contrast when they address "strengths of the study".
Comments on the Quality of English LanguageOk!
Author Response
We thank you and the reviewers for your thoughtful suggestions and insights. The manuscript has benefited from these insightful suggestions.
Reviewer 1
Introduction
1.I suggest that the authors introduce and present what "cerebrovascular disease" is. This is essential so that readers who are not familiar with this disease can better understand the content of your study. Furthermore, the proportion of information between falls and cerebrovascular disease is unequal
“Response: Thank you for highlighting this. We added explanatory text:
Many patients are hospitalized because of cerebrovascular disorders, which refers to disruption in the cerebral blood vessels, resulting in neurological abnormalities such as impaired consciousness and hemiplegia.” Line 37-40
- The authors stated that "Many studies have reported using the Time Up Go test (TUG) to assess fall risk [9]". However, only one study was presented. The literature is vast with examples, please expand the discussion
Response: We added the sentence.
“TUG, which assesses fall risk by walking, may not be possible for stroke patients with decreased walking ability. Therefore, predicting falls based on factors other than motor function is warranted. Although previous studies have mostly evaluated motor function, we believe that factors other than motor function may contribute to fall risk.” line 54-58
- After this statement "Additionally, patients with cerebrovascular disease may not be able to undergo tests such as the TUG owing to hemiplegia or muscle weakness" I suggest complementing with examples of studies that addressed the case, and can support this statement;
Response: We added the references and the sentence.
“Additionally, walking ability is impaired in 80% of patients after stroke.” Line 53-54
- Verma, R.; Arya, K.N.; Sharma, P.; Garg, R.K. Understanding gait control in post-stroke: implications for management. J Bodyw Mov Ther 2012, 16, 14-21.
- It is still necessary to present a strong "justification" for carrying out the study. For example: "To date, there is a gap in studies on ....", or "Author XXX, investigated XXX, Author XXX, evaluated XXX, however, little is still known about XXXX;
Response: Thank you for highlighting this. We added the text:
“Some brain imaging findings are associated with reduced walking ability [13]. Such findings can also be caused by falls, however little is still known about using computed tomography (CT) for fall prediction.” Line 59-61
- I suggest working with a single objective. So far there are two versions: End of the introduction: "This study used brain computed tomography (CT) findings at admission to examine factors associated with falls". And abstract: "This study examined computed tomography findings in patients with cerebrovascular disease and determined factors as predictors for falls".
Response: Thank you for highlighting this. The text for “purpose” has been unified.
Method
- I suggest a better organization of this section, creating subsections:
- a) Presentation of participants;
Response: we have revised the text.
- b) Inclusion and exclusion criteria: detail how the history of falls was collected;
Response: we have revised the text
“The major inclusion criteria were as follows; (1) 20–90 years old, and (2) the patients provided informed consent. The major exclusion criteria were as follows; (1) patients without brain imaging findings, (2) with impaired consciousness, and (3) who were judged by the researchers as unsuitable for participation in the study. Written in-formed consent was obtained from all patients. All study procedures were performed in accordance with the ethical standards of the Institutional Research Committee and adhered to the principles of the Declaration of Helsinki and Ethical Guidelines for Medical and Health Research Involving Human Subjects in Japan.” Line 69-77
- c) Methods used to collect data, specifying the instruments (devices/machines);
Response: We have revised the text:
“All data were collected from medical records.” Line 77
c1) Authors must mention the use of the Mini-Mental State Examination (MMSE) instrument in this subsection;
Response: We have revised the text
“2.2.2. Functional independence measure (FIM)
FIM measurements were performed by evaluating and measuring subjects on a scale of 0-7. FIM can be evaluated from independence to total assistance.
2.2.3. Mini-Mental State Examination (MMSE)
MMSE diagnoses cognitive functions, such as time and place orientation, immediate recall and delayed reproduction, and numeracy, language, and graphic ability. The MMSE contains 11 items, and the perfect score is 30 points. A higher score indicates higher cognitive function. An MMSE score lower than 23 points indicates dementia.
2.2.4. Functional ambulation categories (FAC)
FAC uses a walking path or staircase of about 15 m.
It measured on a scale of 0-6, from inability to walk to independence. It does not matter whether or not prosthetics are used.” Line 96-107
”d) Statistical procedures
* I consider a weakness of the present study that the analyzes were not conducted controlling possible "covariates". It is known that the individuals included have illnesses and certainly consumed medications, as well as having some comorbidities. Therefore, it would be interesting if the regression analyzes presented an unadjusted OR model and another OR adjusted for confounding factors.
Response: Additional analyzes were performed on the unadjusted model. Only the whole is added to the headquarters. The analysis data for cerebral infarction and cerebral hemorrhage are as follows.
Logistic regression analysis results
|
|
OR |
95% CI |
P value |
vif |
|
PVL |
6.3 |
2.16-18.3 |
0.0007 *** |
1.2 |
|
Thalamus |
16.2 |
5.43-47.8 |
0.00000005 *** |
1.10 |
|
Divided into the basal forebrain |
0.000001 |
0-0 |
0.993 |
1 |
|
Lateral ventricle enlargement |
1.21 |
0.44-3.27 |
0.704 |
1.07 |
|
Distant from the lateral ventricle |
1.96 |
0.44-8.6 |
0.37 |
1.01 |
|
Adjacent to the lateral ventricle |
0.61 |
0.21-1.71 |
0.3 |
1.01 |
|
Posterior parietal lobe |
1.12 |
0.19-6.82 |
0.336 |
1.07 |
|
Anterior parietal lobe |
1.1 |
0.299-6.59 |
0.9 |
1.32 |
|
Middle parietal lobe |
2.79 |
0.53-14.7 |
0.22 |
1.10 |
|
Posterior limb of the internal capsule |
2.37 |
0.79-7.08 |
0.121 |
1 |
|
Anterior limb of the internal capsule |
2.04 |
0.52-7.97 |
0.3 |
1.07 |
|
putamen |
0.41 |
0.14-1.22 |
0.11 |
1.07 |
|
|
|
|
|
|
Cerebral hemorrhage
|
|
OR |
95% CI |
P value |
VIF |
|
PVL |
6.3 |
1.66-22.3 |
0.007 ** |
1.04 |
|
Thalamus |
10.1 |
2.93-34.8 |
0.0002 *** |
1.15 |
|
Divided into the basal forebrain |
0.000005 |
0-0 |
0.993 |
1 |
|
Lateral ventricle enlargement |
1.01 |
0.9-1.2 |
0.232 |
1.05 |
|
Distant from the lateral ventricle |
1.45 |
0.55-3.9 |
0.44 |
1.04 |
|
Adjacent to the lateral ventricle |
0.72 |
0.26-1.99 |
0.53 |
1.04 |
|
Posterior parietal lobe |
2.88 |
0.49-15.8 |
0.5 |
1.04 |
|
Anterior parietal lobe |
0.33 |
0.03-1.99 |
0.19 |
1.54 |
|
Middle parietal lobe |
3.49 |
0.88-13.7 |
0.07 |
1.4 |
|
Posterior limb of the internal capsule |
0.44 |
0..9-1.38 |
0.121 |
1.26 |
|
Anterior limb of the internal capsule |
4.44 |
1.12-17.9 |
0.03* |
1.22 |
|
putamen |
0.321 |
0.104-1.02 |
0.06 |
1.11 |
Cerebral infarction
|
|
OR |
95% CI |
P value |
VIF |
|
PVL |
30.3 |
2.06-16.3 |
0.03 * |
1.4 |
|
Thalamus |
250 |
6.43-8200 |
0.002 ** |
1.99 |
|
Divided into the basal forebrain |
0.000001 |
0-0 |
0.993 |
0 |
|
Lateral ventricle enlargement |
4.21 |
0.7-16.2 |
0.134 |
1.3 |
|
Distant from the lateral ventricle |
0.15 |
0.008-8.6 |
0.17 |
1.10 |
|
Adjacent to the lateral ventricle |
6.41 |
1.21-30.7 |
0.03* |
1.2 |
|
Posterior parietal lobe |
1.71 |
0.19-15.8 |
0.636 |
1.03 |
|
Anterior parietal lobe |
6.1 |
0.399-99 |
0.19 |
1.13 |
|
Middle parietal lobe |
0.179 |
0.03-2.07 |
0.22 |
1.17 |
|
Posterior limb of the internal capsule |
9.37 |
1.79-57.8 |
0.121 |
1.65 |
|
Anterior limb of the internal capsule |
2.04 |
0.12-67.97 |
0.3 |
1.04 |
|
putamen |
0.21 |
0.04-1.52 |
0.11 |
1.4 |
Discussion
- I consider that the authors showed ownership in this section;
- On the other hand, the issue of limitations requires a "strong formulation on the "Covariates" point, as well as a failure to present the main clinical characteristics of the participants (i.e., comorbidities, number of medications consumed daily);
Response: Thank you for highlighting this. We added the limitation of this study.
“Therefore, this study did not control for possible "covariates." Additionally, this study did not present the main clinical characteristics of the participants (i.e., comorbidities, number of medications consumed daily). The people included in this study were known to have cerebrovascular disease, were definitely taking medications, and had some comorbidities. We think we should revisit this as we increase the number of cases of cerebrovascular disease [30]. Furthermore, it was necessary to evaluate the relationship between physical functions such as motor function, sensory function, cognitive function, and balance function. Since we also extracted areas where nerve fibers involved in motor function are present, such as the posterior limb of the internal capsule and the area adjacent to the lateral ventricle, it is possible that purely motor function is affected. We believe that physical findings remain important, as not all clinical professionals have the financial means to apply imaging tests. Hospitals and facilities are taking several steps to prevent falls, such as bed rails or restraints.” line 248-261
- Zhou, S.; Jia, B.; Kong, J.; Zhang, X.; Lei, L.; Tao, Z.; Ma, L.; Xiang, Q.; Zhou, Y.; Cui, Y. Drug-induced fall risk in older patients: A pharmacovigilance study of FDA adverse event reporting system database. Front Pharmacol 2022, 13, 1044744.
- I suggest, presenting strengths of the study;
Response: We add the sentence at the beginning of the discussion.
“This study suggested that fall risk can be easily assessed using CT images without requiring patients to walk. We believe that if we can understand the risk of falling be-fore a physical examination, we may be able to better prevent falls.” Line 177-179
- I suggest presenting suggestions for future investigations: directions based on the results found.
Response: Thank you for highlighting this. We added explanatory text “Hospitals and facilities are taking several steps to prevent falls, such as bed rails or restraints. We believe that by adding predictions using CT images to multi-faceted verification, we can further improve the accuracy of fall risk estimation.” Line 260-262
Conclusion
- I consider it clear and consistent.
- point for reflection: The authors stated the following "Image
based prediction of falls will facilitate risk management and effective interventions during rehabilitation". However, it is known that not all clinical professionals have the financial means to apply imaging tests to detect risk of falls in their patients. Therefore, I suggest that this point be presented in contrast when they address "strengths of the study".
Response: Thank you for highlighting this. “We believe that physical findings remain important, as not all clinical professionals have the financial means to apply imaging tests.” Line 258-260
I also rewrote the Conclusion text.
“Image based prediction of falls will facilitate risk management and effective interventions during rehabilitation. The effect may become more accurate by combining it with physical functions.” line 265-267
Reviewer 2 Report
Comments and Suggestions for Authors
Dear Authors,
See some suggestions to improve the manuscript.
Abstract:
1. Please add some values about the results
Introduction
1. Can authors explain a litle more about "Brain imaging findings predict physical function but not falls".
2. There is some evidence about falls by sex? Please explore it a litle more in introduction.
3. in line 46 add more references please
4. this manuscript is about predicting falls. can you add more information about the risk of falls and the contribution to this research in quality of life in older people?
5. please add the hypotheses of the study in the final of the introduction
Methods
1. the inclusion criteria was with people around 20 and 90 years, but the final participants were about 70 years. whats happened with the young adults? You think that your study are focus on older people? If yes, maybe you can add it in the title
2. how they were divided in Fall and not Non-fall? We say the reference 4 in line 71, but can you add some information to clarify it?
Discussion
1. between line 194 and 194 "Based on these results, we believe it is meaningful to add the evaluation of brain imaging findings to the physical function evaluation of the lower extremities when evaluating falls." if is possible add information about the importance of physical exercise to prevent and decrease the risk of falls.
2. During all the discussion there is a clinical explanation, but you can add more the consequences about it in the performance of daily activities? Pratical consequences concerning the brain modifications.
3. In conclusions authors refered "physical function evaluations." can you add some examples, above TUG, what professional can do more?
Author Response
We thank you and the reviewers for your thoughtful suggestions and insights. The manuscript has benefited from these insightful suggestions.
Reviewer 2
Abstract:
- Please add some values about the results
 Response: Thank you for highlighting this. I added the value.
Introduction
1.Can authors explain a litle more about "Brain imaging findings predict physical function but not falls".
Response: We add the sentence “Some brain imaging findings are associated with reduced walking ability [13]. Such findings can also be caused by falls, however little is still known about using computed tomography (CT) for fall prediction.” Line 59-61
- There is some evidence about falls by sex? Please explore it a litle more in introduction.
Response: Thank you for your comment. We considered gender differences, but as shown in the figure below, we did not find any gender differences in this study.
Proportion of gender with cerebrovascular accident
- in line 46 add more references please
We added the two references.
・Kojima G, Masud T, Kendrick D, Morris R, Gawler S, Treml J, Iliffe S. Does the timed up and go test predict future falls among British community-dwelling older people? Prospective cohort study nested within a randomised controlled trial. BMC Geriatr. 2015 Apr 3;15:38.
・Beauchet O, Fantino B, Allali G, Muir SW, Montero-Odasso M, Annweiler C. Timed Up and Go test and risk of falls in older adults: a systematic review. J Nutr Health Aging. 2011 Dec;15(10):933-8.
- this manuscript is about predicting falls. can you add more information about the risk of falls and the contribution to this research in quality of life in older people?
Response: We added the sentence “The purpose of this research is to identify a method that can be used to easily predict falls. Falls worsen the quality of life of older people and increase the risk of becoming bedridden [14].” Line 61-63
- please add the hypotheses of the study in the final of the introduction
Response: Thank you for your comment. “We hypothesized that the cause of falls can be predicted from brain imaging findings. This study examined CT findings in patients with cerebrovascular disease and deter-mined factors as predictors for falls.” Line 63-65
Methods
- the inclusion criteria was with people around 20 and 90 years, but the final participants were about 70 years. whats happened with the young adults? You think that your study are focus on older people? If yes, maybe you can add it in the title
Response: Most of the participants this time are over 60 years old, but there are others as well. For this reason, I did not include "older adult" in the title this time. I also added range to Table1
|
|
Fall group (n=43) |
Non-fall group (n=95) |
P value |
|
Age |
73.6±10.0 |
73.9±9.55 |
0.84 |
|
|
52-90(range) |
47-88(range) |
|
|
Duration from onset to CT |
36.1±22.9 |
38.2±24.4 |
0.63 |
|
|
0-108(range) |
0-110(range) |
|
|
MMSE |
14.5±9.1 |
14.6±11.2 |
0.92 |
|
|
0-28(range) |
0-30((range) |
|
|
FIM |
34.9±12.9 |
43.4±24.4 |
0.008 ** |
|
|
21-119(range) |
18-123(range) |
|
|
Total number of CT findings |
2.67±1.42 |
1.89±1.47 |
0.004 ** |
|
|
0-4(range) |
0-6(range) |
|
|
FAC |
2.0±1.66 |
2.27±2.0 |
0.40 |
|
|
1-5(range9 |
0-5(range) |
|
- how they were divided in Fall and not Non-fall? We say the reference 4 in line 71, but can you add some information to clarify it?
Response: Thank you for your comment. We added the sentence “Gibson’s definition is as follows: “[a fall] is the inadvertent fall of a person onto the same or lower plane, not due to external force caused by another person, loss of consciousness, sudden onset of paralysis, epileptic seizure, etc.”
78-80
Discussion
- between line 194 and 194 "Based on these results, we believe it is meaningful to add the evaluation of brain imaging findings to the physical function evaluation of the lower extremities when evaluating falls." if is possible add information about the importance of physical exercise to prevent and decrease the risk of falls.
- In conclusions authors refered "physical function evaluations." can you add some examples, above TUG, what professional can do more?
Response: I apologize for any confusion. The revised text is below with addition of references.
“As reported in previous studies, physical function is important for preventing falls in older adults [26]. Although evaluations for fall risk in patients with cerebral vascular disease, such as TUG, Berg Balance Scale score (BBS), Stroke Assessment of Fall Risk (SAFR), and Morse Fall Scale are used in many hospitals, these evaluations do not predict falls on their own [27-29].” 238-242
・Ikegami S, Takahashi J, Uehara M, Tokida R, Nishimura H, Sakai A, Kato H. Physical performance reflects cognitive function, fall risk, and quality of life in community-dwelling older people. Sci Rep. 2019 Aug 22;9(1):12242.
- During all the discussion there is a clinical explanation, but you can add more the consequences about it in the performance of daily activities? Pratical consequences concerning the brain modifications.
Response: Thank you for highlighting this. It is difficult to explain each body part and daily life activities in this paper. However, since there were many areas of failure related to alertness, I added a sentence.
“The majority of the brain that was found to be associated with falls in this study is the part that affects alertness and cognitive function, and we believe that alertness and cognitive function influenced falls.” 244-247
Round 2
Reviewer 1 Report
Comments and Suggestions for Authors
Dear all, I consider that the adjustments have been made, and that the text is now ready to be published.
Sincerely!